# Potential Valorization of Banana Production Waste in Developing Countries: Bio-Engineering Aspects

Robert Waraczewski and Bartosz G. Sołowiej *

Department of Dairy Technology and Functional Foods, Faculty of Food Sciences and Biotechnology, University of Life Sciences in Lublin, Skromna 8 St., 20-704 Lublin, Poland; robert.waraczewski@up.lublin.pl
* Correspondence: bartosz.solowiej@up.lublin.pl

**Abstract:** Plant food production generates a lot of by-products (BPs). These BPs are majorly discarded into the environment, polluting it, or into landfills where they just decompose, providing no benefit and taking up storage space, causing financial costs. These plant BPs are biodegradable, but reusing them may provide a better outcome and profit. The vast majority of plant-based food BPs are polysaccharide polymers like gums, lignin, cellulose, and their derivatives. It is possible to utilize plant food production waste, like banana peels, leaves, pseudostems, and inflorescences, to produce bioethanol, single-cell protein, cellulase, citric acid, lactic acid, amylase, cosmetics, fodder additives, fertilizers, biodegradable fibers, sanitary pads, bio-films, pulp and paper, natural fiber-based composites, bio-sorbents, bio-plastic, and bio-electricity in the agro-industry, pharmaceutical, bio-medical, and bio-engineering fields. Moreover, the use of banana BPs seems to be a way of dealing with many issues in underdeveloped countries, providing a clean and ecological solution. The suggested idea might not only reduce the use of plastic but also mitigate waste pollution.

**Keywords:** natural fibers; environmental sustainability; banana; polysaccharides; bio-based economy





## 1. Introduction

Bananas are the most cultivated fruit worldwide, and their production is projected to grow further. The main reason for this is their versatile usage in cuisine in banana producing countries and the high demand for affordable fruits for import. On the other hand, there are some factors limiting the production and commerce, like product shortages caused by harsh weather; high fertilizer costs in 2022 and early 2023; fruit losses; prevention and fighting costs regarding plant diseases, mainly Banana Fusarium Wilt Tropical Race 4; and losses and additional costs regarding the prevention of illegal substances being placed inside banana containers exported from Latin America [1].

Approximately 135 million tons were produced worldwide in 2022, exceeding watermelons (100 million tons), apples (96 million tons), oranges (76 million tons), and grapes (75 million tons) [2]. India leads in the world banana production; however, only about 0.04% are exported. The rest are consumed locally. The reason for this is the aggressive competitiveness of other exporters and their fruit's high quality, which India cannot always meet. Moreover, banana production in India is not reliable: rapid changes in production have been reported. Also, the population of India is rising quicker than banana production growth [3]. The leading exporting countries are Ecuador, the Philippines, Costa Rica, Guatemala, Colombia, and the Dominican Republic, which account for about 60% of the world export [1,4].

The banana palm is a perennial herb. After bearing fruit, its above-ground parts—the pseudostem, leaves, inflorescence, and fruit stalks—die, leaving space for suckers to grow as a new plant. This cycle can theoretically last forever. Considering the amount of harvested fruit, for each ton of bananas produced, 2–4 times the weight of banana by-products (BPs) is created, depending on the variety, leaving a significant load of BPs. This

is about 220 tons of BPs per hectare per year. Some of these BPs like pseudostems and leaves may be left to rot, making a natural fertilizer. The young shoots, pseudostem piths, and inflorescence may be consumed by the indigenous people of Southeast Asia and the Indo-Malaysian Region; however, this is rather uncommon due to the BPs unpleasant taste [4–6]. In developing countries with a human development index < 0.8, [7] like India, bananas are a main component in diets but also a main source of income and essential cultural and rite paraphernalia [8]. A map of developing countries and the countries with notable banana production has been developed (Figure 1). The benefits of banana peel usage have been suggested by many authors, but no practical applications have been provided [9]. Many natural polysaccharide by-products in general have applications, e.g., antioxidant, antitumor, immunomodulatory, antimicrobial, antiulcer, and hypoglycemic activities and as prebiotics [10] and substrates for bio-fuel [11]. Starch, found in bananas, mainly unripe, may be a source of dietary fiber. As the fruit ripens, the starch breaks down into simple sugars, which may be used for fermentation [12]. Moreover, numerous trials have concluded that there are no harmful phytochemicals in banana fruit and BPs; therefore, the utilization of the presented fruit and BPs seems safe, cheap, beneficial, and feasible [4].

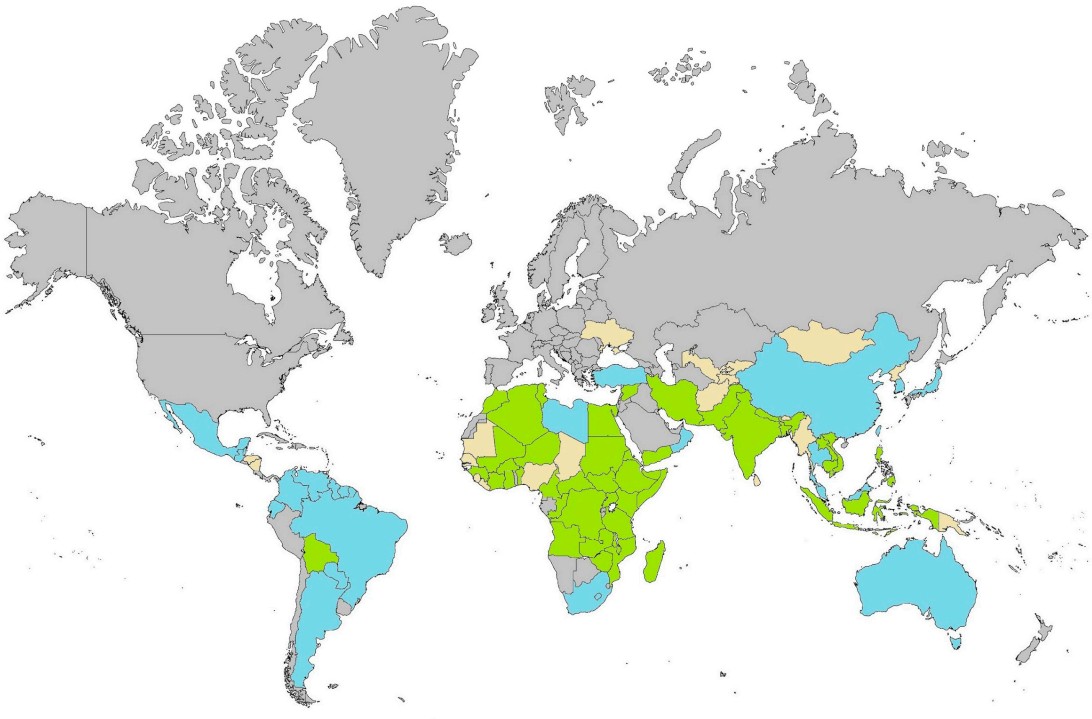

**Figure 1.** Developing countries according to the World Population Review [7] (yellow color), the countries of Asia, Africa, Australia, and Latin and South America with notable banana production (blue color), and countries which meet both criteria (green color). Own figure according to Atlas Big [13].

## 2. Banana By-Product Utilization

The banana plant's pseudostem and leaves are mostly discarded as waste. According to Sellin et al. [6], for every 1 ton of bananas harvested, 1.5 tons of leaves and 2.5 tons of pseudostem are produced. However, these BPs can be utilized in various ways. The composition of banana BPs is presented in Table 1.

**Table 1.** Selected chemical compound composition of different banana BPs. m expressed as % molar proportion; % expressed as % ash. Own table according to Cordeiro et al. [14] and Oliveira et al. [15].

| Compound | Inflorescence | Leaf Sheaths | Leaf Blade | Petioles/mid rib | Pseudostem | Rachis |
|---|---|---|---|---|---|---|
| Arabinose | 5.1 m | 7.5 m | 15.5 m | 4.9 m | 9.1 m | 4.1 m |
| Cellulose | 15.7 m | 37.3 m | 20.4 | 31.0 m | 34–40 m | 31.0 m |
| Galactose | 2.9 m | 2.2 m | 3.8 m | 1.1 m | 2.5 m | 1.7 m |
| Glucose | 79.8 m | 74.2 m | 60 m | 68.1 m | 74.0 m | 31.8 m |
| Holocellulose | 20.3 m | 49.7 m | 32.1 m | 62.7 m | 60–65 m | 37.9 m |
| Lignin | 10.7 m | 13.3 m | 24.3 m | 18.0 m | 12.0 m | 10.5 m |
| Mannose | 2.2 m | 1.5 m | 2.3 m | 1.5 m | 1.3 m | 2.9 m |
| Rhamnose | 0.7 m | 0.8 m | 0.9 m | 0.8 m | - | 0.7 m |
| Xylose | 9.3 m | 13.8 m | 17.5 m | 23.6 m | 13.1 m | 14.0 m |
| Ash | 26.1% | 19.0% | 19.4% | 11.6% | 14.0% | 26.8% |
| Calcium | 0.6% | 5.5% | 8.0% | 32.3% | 7.5% | 0.6% |
| Magnesium | 0.5% | 1.9% | 1.1% | 2.9% | 4.3% | 0.3% |
| Phosphorous | 0.7% | 0.9% | 0.7% | 0.7% | 2.2% | 1.7% |
| Potassium | 23.1% | 21.4% | 11.6% | 9.4% | 33.4% | 28.0% |
| Silicon | 7.8% | 2.7% | 24.9% | 7.0% | 2.7% | 1.2% |
| Pentosans | 8.0 m | 12.4 m | 12.1 m | 16.2 m | - | 8.3 m |
| Proteins | 3.2 m | 1.9 m | 8.3 m | 1.6 m | - | 2.0 m |
| Starch | 26.3 m | 8.4 | 1.1 m | 0.4 m | - | 1.4 m |

The pith of banana plants' pseudostem and unripe discarded fruit are a good source of starch, pectin, and cellulose. Starch isolated from banana BPs is notably resistant to the influence of heat and amylase. Its water solubility, retrogradation, and swelling properties are low; thus, banana BPs surpass modified and unmodified corn starch slightly regarding physicochemical properties [16]. Pectin obtained from banana peels show a slightly lower methoxyl composition and gelling quality compared to peels from citruses like pomelo or lime. Despite the slightly lower gelling quality of pectin, it can also be used as a dietary fiber source, making banana peels a worthy source of this dietary compound. On the other hand, banana peels contain more dietary fiber than plantain peels, and both of these peels can produce a similar or higher amount of dietary fiber compared to other plants. Moreover, the extraction of pectin and dietary fiber from banana BPs can reduce the need for importing these substances into banana producing countries. Pectin extracted from bananas consists of neutral sugars (galactose, arabinose, rhamnose) and galacturonic acid. The quality of the extracted pectin depends on the method used and the ripeness of the fruit [4,17–19]. Pectin extraction from banana peels is presented in Figures 2 and 3.

Khamsucharit et al. [20] used the mentioned method to extract pectin from the peels of five banana varieties within three genomic groups, namely Kluai Khai and Kluai Leb Mu Nang of the Musa (AA) group, Kluai Hom Thong of the Musa (AAA) group, and Kluai Nam Wa and Kluai Hin of the Musa (ABB) group. The pectin yield varied at about 16.24%, 24.08%, 17.31%, 15.89%, and 19.48%, respectively, which is similar or even higher compared to the pectin extracted from the most commonly used substrates—citrus peels and apple pomace (19.25% and 10.91%, respectively). Moreover, all banana peel pectin was characterized as high-methoxyl type, being similar to citrus peel and apple pomace pectin. Pectin extracted from the Kluai Nam Wa variety was the purest, meeting the criteria for a food additive.

Banana peel may also be utilized in carboxymethyl cellulose (CMC) production—a popular and versatile thickening agent, texture modifier, and emulsifier used in the food, cosmetic, pharmaceutic, personal and health care, paint, paper processing and packaging industries. The process of extracting CMC from banana peels is presented in Figures 4 and 5 [21].

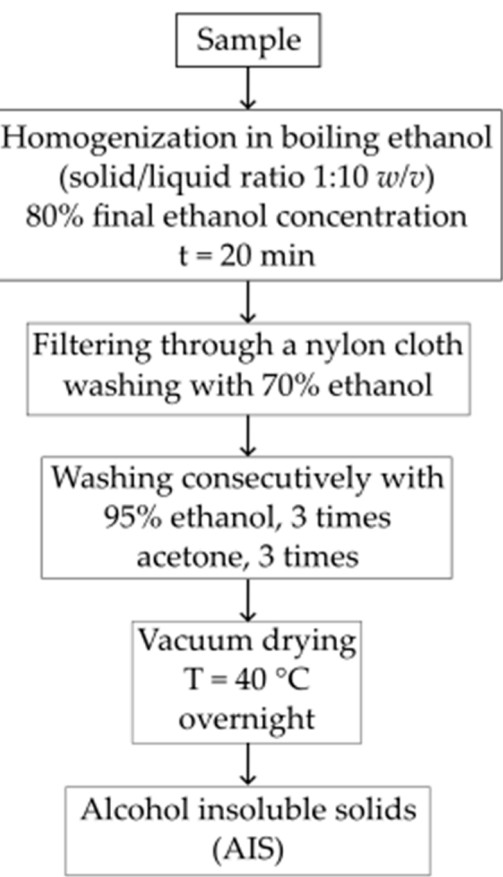

**Figure 2.** Isolation of alcohol insoluble solids (AISs) from a sample, e.g., banana peels. AISs are required for further pectin extraction. Own figure according to Khamsucharit et al. [20].

The yield of raw banana peel cellulose was about 14% (% dry basis). The raw banana peel cellulose was characterized by powder grains bigger than banana peel CMC gains. The grains were flake-shaped and of a brown-yellow color. In the case of CMC, the extraction time of 4 h provided the highest yield—about 152.65%. A shorter extraction led to a lower yield because of an incomplete process, and a longer extraction caused the degradation of banana peel CMC, also causing a lower yield [21].

Unripe, green bananas are most often discarded as waste or become animal feed. Such an approach is not economical, since about 70–80% of banana (without skin) dry weight is starch, which can be extracted for further processing in the food, pharmaceutical, or beauty industries. One of the most important factors defining starch content is the banana plant's genomic group. Generally, plants with a high proportion of *M. balbisiana* (B genes) accumulate more starch than plants with a high proportion of *M. acuminata* (A genes). Starch is a valuable substance in cosmetics production, providing a smooth feel to the products and acting as a stabilizer, thickener, or a talcum substitute. Thanyapanich et al. [22] evaluated the percentage yield of starch from *Musa acuminata* (Musa AAA; Hom Khieo) and *Musa sapientum* L. (Musa ABB; Namwa) unripe banana plant varieties. The yield was 16.88% and 22.73%, respectively. The amylose content was 24.99% and 26.23%, respectively. Starches from both banana varieties showed a B-type crystalline structure, and their gelatinization temperature was at about 77 °C. For cosmetic purposes, both starches provided a smooth feel and acceptable spreading properties. Moreover, starch may be used as a substitute for talcum without any sensorial changes if the starch concentration does not exceed 15% (*w/w*). The process of starch extraction from unripe banana fruit is presented in Figure 6. The unripe banana is firstly cut into cubes, then blended with a sodium sulfite solution for 10 min. The cubes remain immersed at a 1:2 ratio with a sodium sulfite solution. Next, the mixture is filtered and centrifuged to obtain precipitated starch, which needs drying in dry

conditions. The dry starch is crushed into a powder with a mortar and pestle and sieved until the final product is finished. Banana BPs may also be converted into biomass, which may be further processed into bioethanol used in the cosmetics industry [19,23].

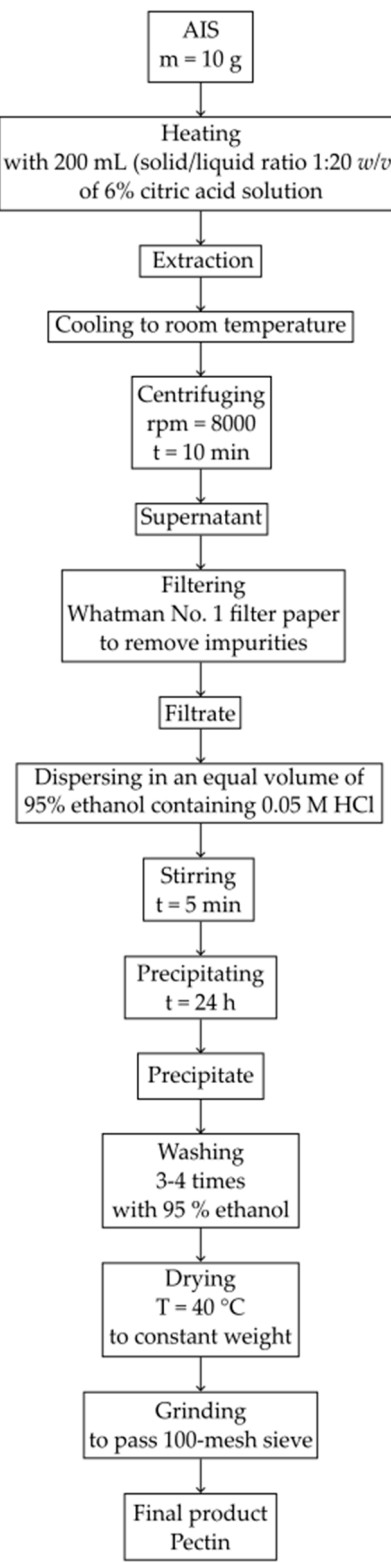

**Figure 3.** Extraction of pectin from AISs (e.g., banana peel AISs). Own figure according to Khamsucharit et al. [20].

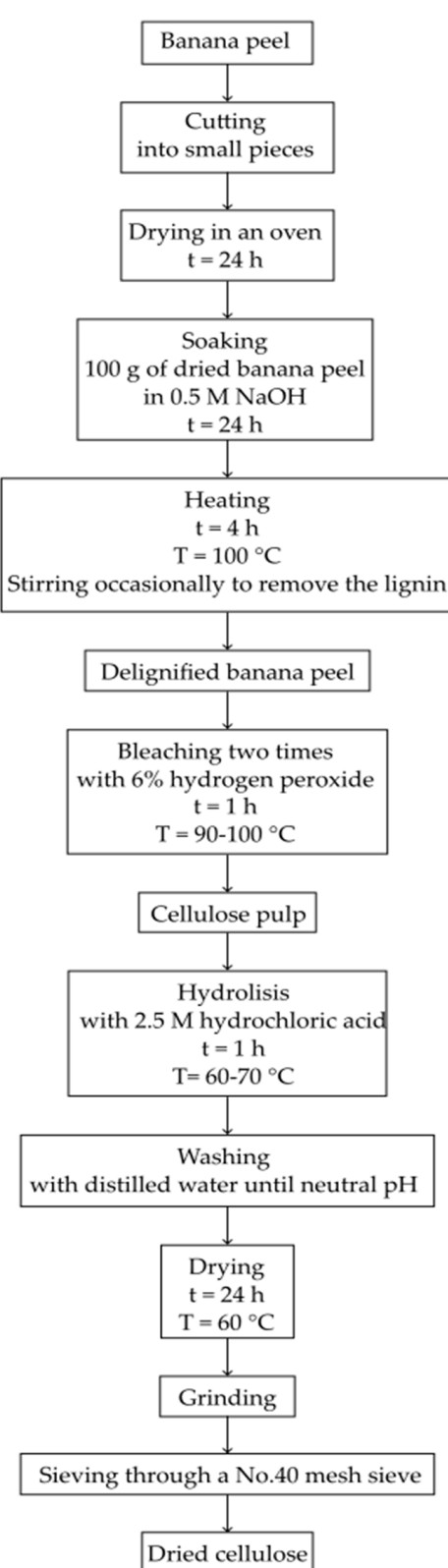

**Figure 4.** Extraction of dried cellulose from banana peels. Dried cellulose in an essential substrate for banana peel CMC extraction. Own figure according to Suchaiya et al. [21].

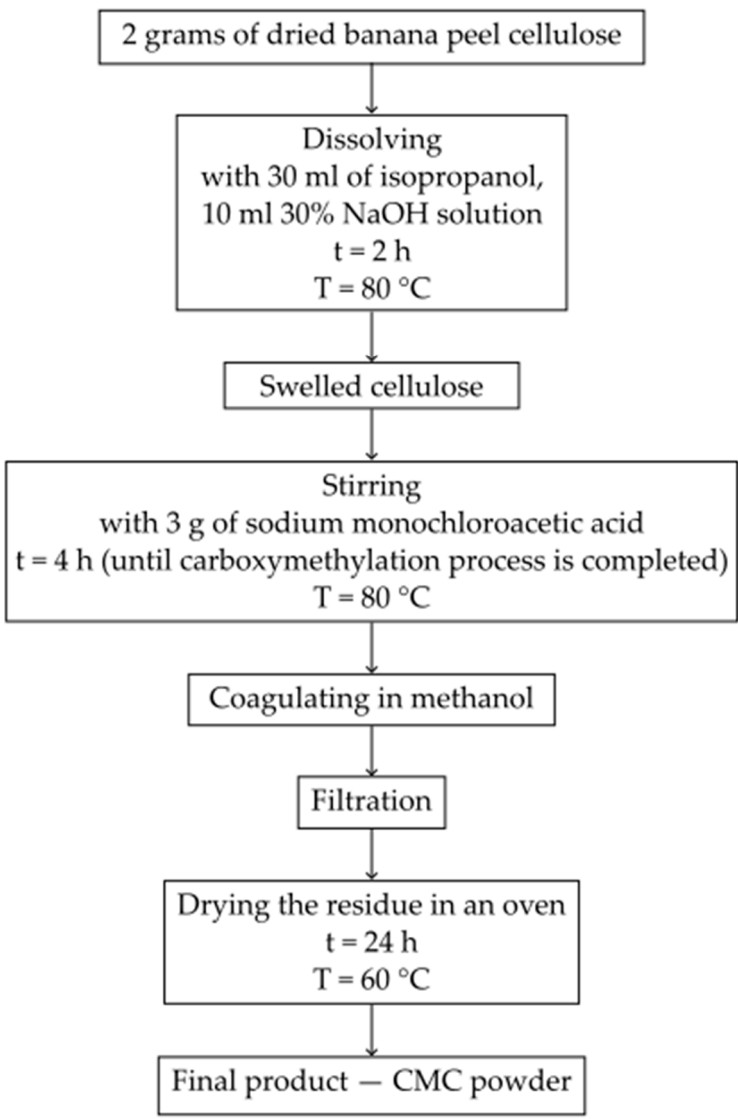

**Figure 5.** Extraction of CMC from dried cellulose of banana peel. Own figure according to Suchaiya et al. [21].

Bananas, plantains, and their products are significant to the people of Colombia. These fruits and products are widely used raw because of their high starch and potassium contents; moreover, they are an important ingredient in soups, fried slices, traditional dishes, and sweet or sour snacks made for export and local consumption. Such extensive consumption of bananas and plantains causes a large amount of BPs like peels. These BPs may be used as a source of starch. There are two main methods of extracting starch from plantain pulp—dry and wet—with an extraction yield of about 49.62% and 56.76%, respectively. One way of wet extracting starch from banana BPs is a method suggested by Hernandez-Carmona et al. [24]. The procedure for extracting starch from green plantain (*Musa paradisiaca*) peels is shown in Figure 7. This starch may be used in the paper, textile, pharmaceutical (as an excipient), adhesive, food, and polymer industries and as a coagulant for water filtering.

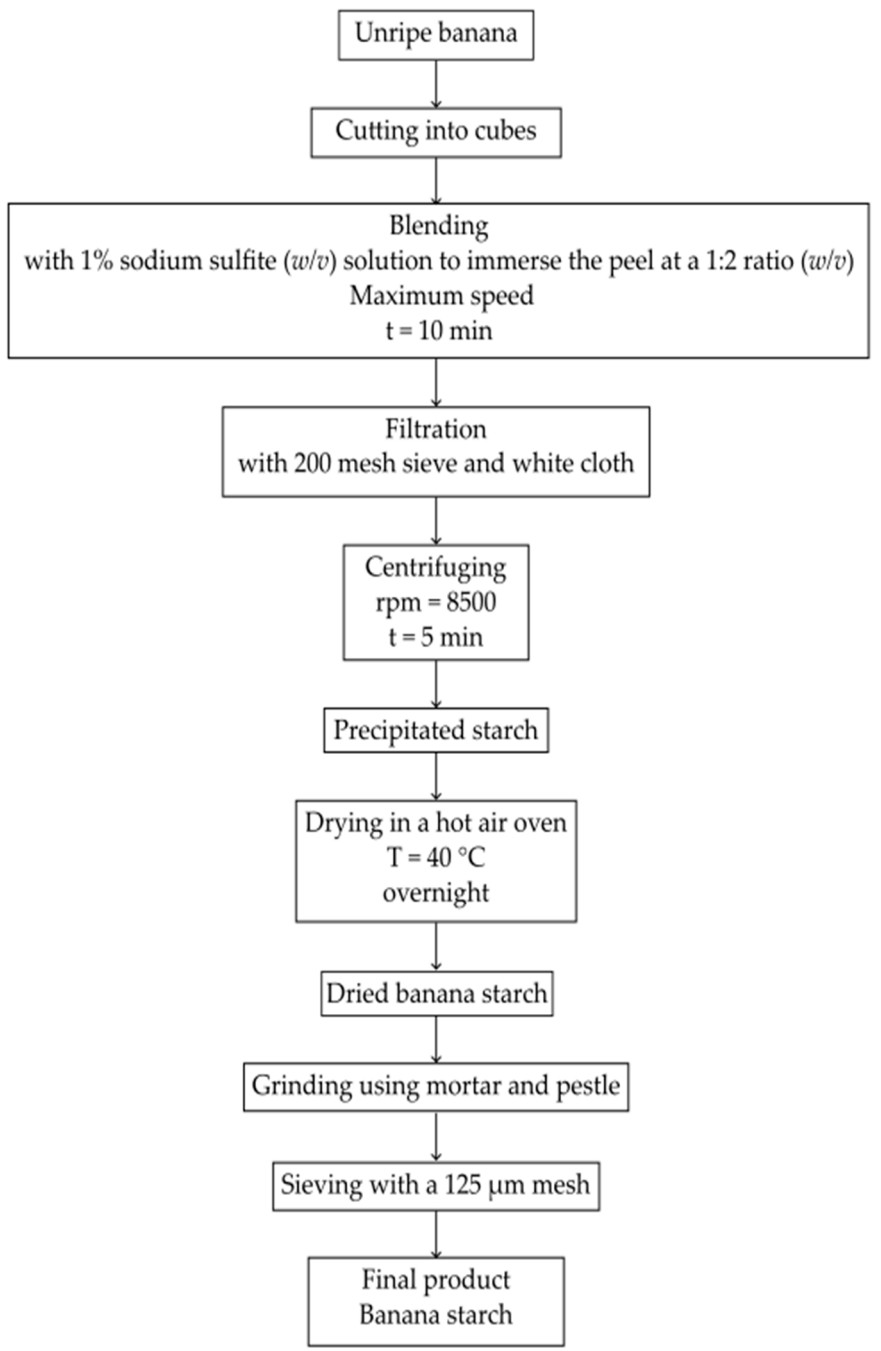

**Figure 6.** Extraction of starch from unripe banana. Own figure based on Thanyapanich et al. [22].

Raw, unripe plantain fruit contains about 70–80% starch (dry basis), while the peel alone contains up to 50%. As the plant matures, the starch present in the fruit breaks down to simpler sugars. The sugar content in ripe plantains and bananas may reach up to 21%. The technology suggested by Hernandez-Carmona et al. [24] provided a yield of up to 48.5%, average 29%, (of dry basis) depending mainly on the antioxidant concentration—the highest ascorbic acid concentration led to the highest yield. The reason for this is the stabilizing effect of antioxidants on the starch—ascorbic acid prevents the breaking down of the starch to simple sugars. The purity of the starch was about 57.52 to 69.9%.

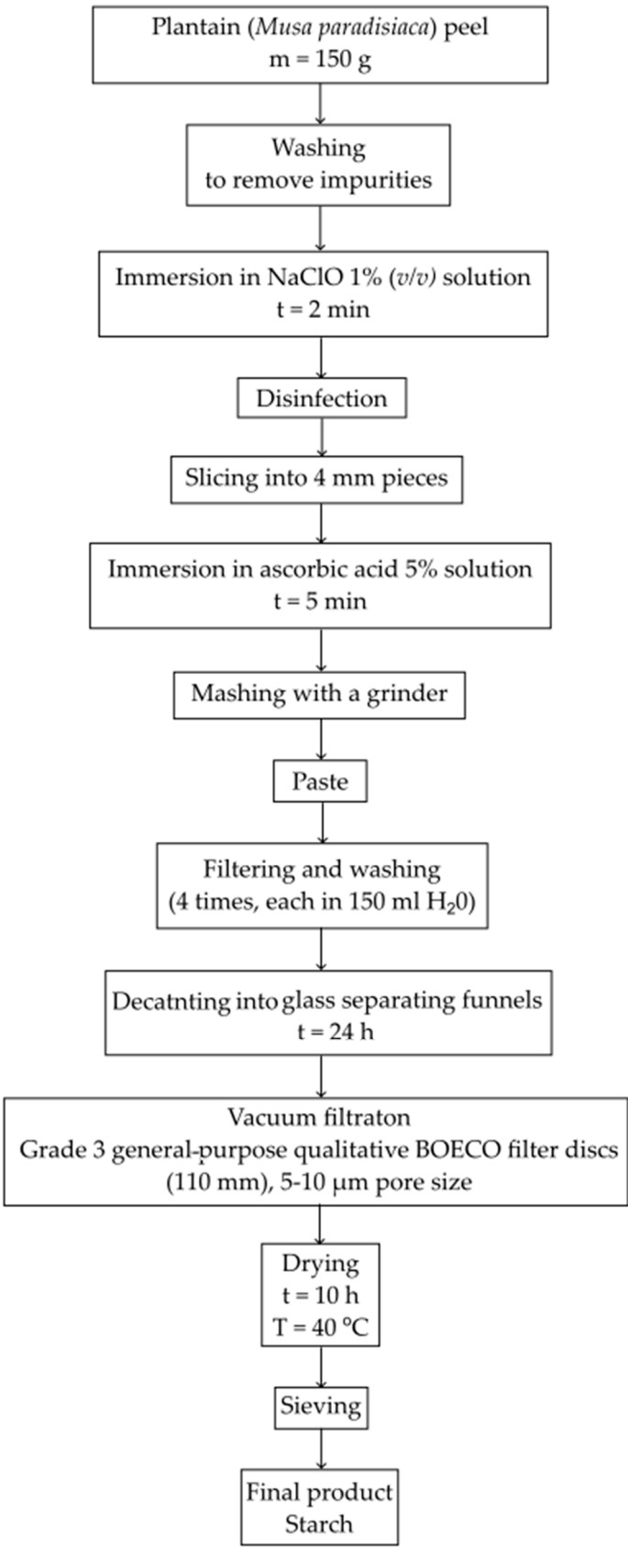

**Figure 7.** Wet extraction of starch from plantain peels. Own figure according to Hernandez-Carmona et al. [24].

Another noteworthy banana BP is its flower. It contains plenty of proteins (10–12.5% dry weight), dietary fiber, vitamins (like vitamin C), flavonoids (mainly quercetin), and biologically active compounds, e.g., tannin and α-tocopherol. The people of Sri Lanka, Malaysia, the Philippines, and Indonesia consume banana blossoms in the form of curry, in salads, or boiled. Moreover, they can be incorporated into canned food, pickles, and dehydrated vegetables [25].

Inflorescences contain enough starch to make it worth obtaining and can be used in the pharmaceutical and food industries [15].

Due to the presence of colorful anthocyanins, flower buds can be used to produce cheap, moderately stable food colorings. Moreover, anthocyanins are proven to be beneficial for human health [4].

According to Tin et al. [26], male banana flowers contain epigallocatechin and its derivatives, which are classified as nutraceutical compounds. Epigallocatechin-3-gallate has been confirmed to prevent diabetes, cancer, cardiac disease, lung disease, and neuro degenerative disease [27].

Banana flower and chicken meat can be mixed together to create a new product with an enhanced color, aroma, taste, texture, overall desirability, and pro-health effect due to the presence of antioxidants [28]. Banana flower extracts can be used to enrich pork burgers and sausages, slowing lipid oxidation [29,30].

About one-third of the weight of a whole banana fruit is the peel, which is commonly disposed of as general waste [31] or into landfills [32]. The composition of banana peel is presented in Table 2.

**Table 2.** Banana peel composition of dessert bananas (AAA and AAB variety); % (*w/w*), * dry weight. Own table based on Archibald [33]; Lustre et al. [34]; Adisa and Okey [35]; and Anhwange et al. [36].

| Compound | Peel |
|---|---|
| Moisture | 83.50% |
| Total sugar | 29% |
| Cellulose | 8.40% |
| Fructose | 6.20% |
| Sucrose | 2.60% |
| Glucose | 2.40% |
| Proteins | 1.80% |
| Fat | 1.70% |
| Starch | 1.20% |
| Maltose | 0% |
| Potassium (K) | 78.10 ± 6.580 * mg/100 g |
| Manganese (Mn) | 76.20 ± 0.001 mg/100 g |
| Sodium (Na) | 24.30 ± 0.120 mg/100 g |
| Calcium (Ca) | 19.20 ± 0.001 mg/100 g |
| Iron (Fe) | 0.61 ± 0.001 mg/100 g |
| Rubidium (Rb) | 0.21 ± 0.050 mg/100 g |
| Bromine (Br) | 0.04 ± 0.001 mg/100 g |
| Strontium (Sr) | 0.03 ± 0.010 mg/100 g |
| Niobium (Nb) | 0.02 ± 0.001 mg/100 g |
| Zirconium (Zr) | 0.02 ± 0.001 mg/100 g |

### 2.1. Utilities

The pseudostem is used to make knit fabric [37] and various types of paper, e.g., paper boards. Leaves are widely used as a material for baskets, mats, food wrappers, tablecloths, and tableware and by rural people as an umbrella. To prevent bats and birds from feeding on the banana fruit, old leaf coverings are utilized. Dried leaves are used as a substrate for subsoil for oyster mushroom cultivation. In India, the leaves are essential for traditional rituals and rites [8].

Fibers may be used to produce water purifiers [8], which seem especially important in the developing countries of Africa (mainly Chad, Niger, the Central African Republic, and South Sudan) and Asia (mainly India) [38], where waterborne diseases are common [39]. The leaf blades, floral stalks, leaf sheaths, and rachis contain a lot of ash and thus can be used as an alkali ingredient in soap production [40]. The pseudostem pith is used as an absorbent for textile dyes present in wastewater [41]. This may be especially useful in Bangladesh, where textile manufacturing is notable, and textile dye pollution is present in water reservoirs [42]. Modified pseudostem may be used as an efficient $Pb^{2+}$ and $Cd^{2+}$ sorbent [43].

It is possible to use pseudostem fibers to manufacture biodegradable nanocellulose plastic and films [44] or to strengthen thermoplastic composites [45]. This seems especially important in Tanzania, Vietnam, South Africa, India, Costa Rica, China, and Chile, where marine plastic pollution is a confirmed issue [46].

Fibers isolated from banana pseudostem are used as a raw material in traditional clothes and handicraft production [47].

The inner part of banana pseudostem is consumed with salt and spices in India after heat treatment in water [48].

## 2.2. Animal Feed

The leaf meal, fresh leaves, fresh foliage, dried leaves, and dried pseudostem of bananas and plantains are used as a fortifying compound or as a partial replacement for more expensive ingredients in animal feed [49]. Leaves have a low partition factor, high ATP, and high microbial biomass, thus making them a suitable feed for ruminants [50]. Moreover, banana root intake contributes to lower coccidiosis infections in rabbits, reducing the fecal egg count after 14 days of treatment [51]. Such treatment may find use in countries like Kenya [52] and Indonesia where rabbits are bred and coccidiosis infections are common [53].

Due to the high amount of protein, lipid, carbohydrate, fiber, and essential minerals like potassium, sodium, calcium, iron, and manganese, banana peels are a valuable material for fodder production [36]. Using banana peel subsoil to grow micro-fungi obtains biomass enriched with protein and fatty acids in a solid mixture [54]. Such an enrichment allows for the production of high-nutritional feed from low-quality materials [55].

Banana peels may be used to make livestock feed, replacing more expensive ingredients like soybean or cassava [36,56].

## 2.3. Bioactive Compounds

Banana pseudostem, leaves, and roots contain enzymes such as phosphorylase, γ-amylase, phospho-hexo-isomerase, acid and alkaline invertase, sucrose synthase, sucrose phosphate synthase, and acid and alkaline phosphatase [57]. Starch phosphorylase has industrial importance. It is used to produce modified starch, novel glucans, and glucose-1-phosphate (Glc-1-P). Glc-1-P is especially noteworthy because of its applications—it is utilized for glucan, glucuronic acid, and trehalose synthesis as a substrate and has medical purposes including cardiotherapy since it is a cytostatic compound and an ingredient in antibacterial, anti-inflammatory, and antitumor agents. Glucans and their modified forms made of Glc-1-P are used in raw materials in starch processing industries, food and drink manufacturing, food additives, and biodegradable plastics [58]. The process of phosphorylase (fractions containing phosphorylase activity) extraction from unripe banana is complex and requires many precise steps; thus, it is explained in Figure 8.

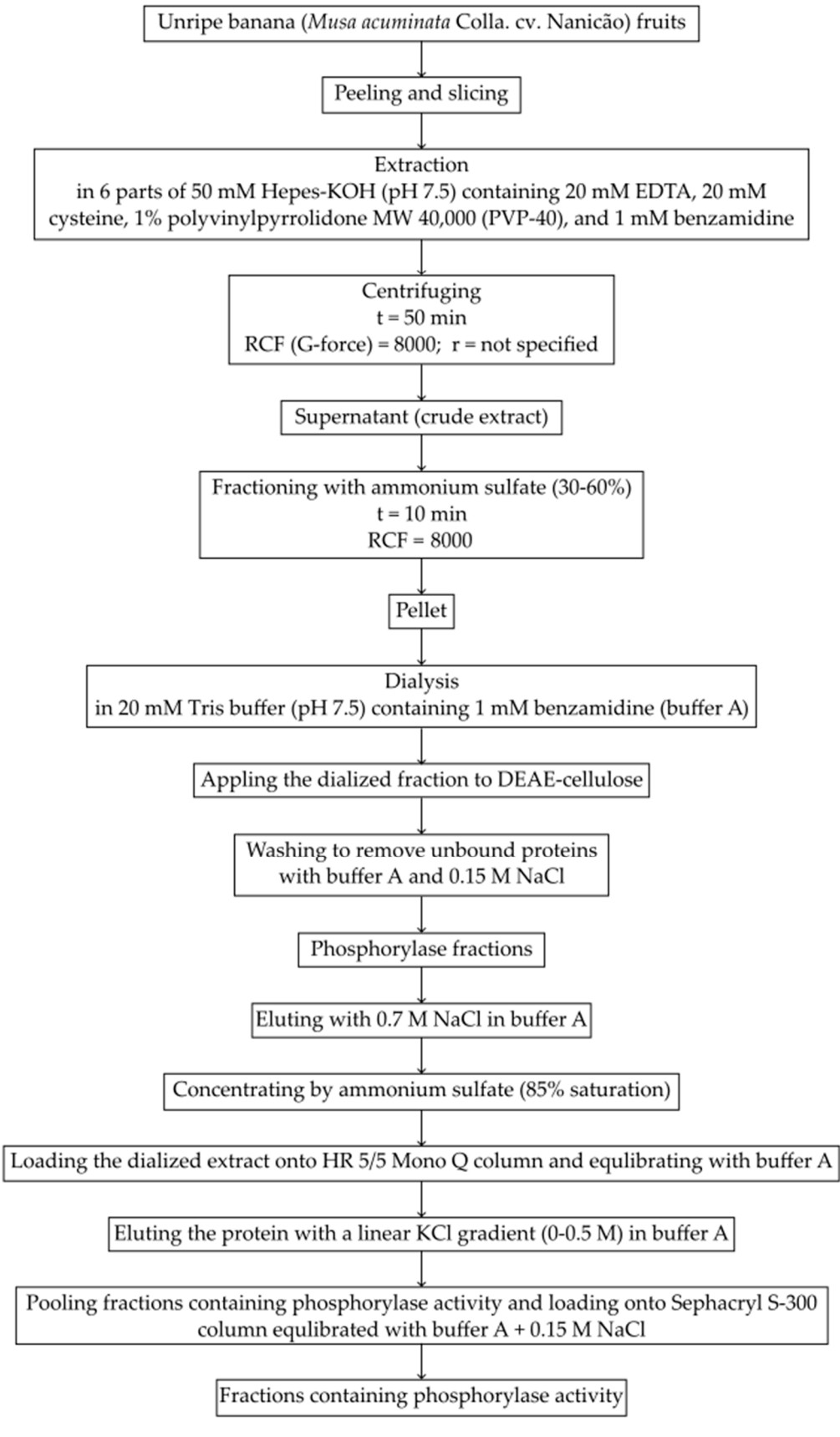

**Figure 8.** Extraction of phosphorylase from unripe banana (*Musa acuminata* Colla. cv. Nanicão) fruits. Own figure based on da Mota et al. [59].

Da Mota et al. [59] extracted two different forms of phosphorylase from unripe banana (*Musa acuminata* Colla. cv. Nanicão) fruits—phosphorylase I and phosphorylase II. The characteristics of both compounds are presented in Table 3.

**Table 3.** Characteristics of phosphorylase I and II obtained from unripe banana (*Musa acuminata* Colla. cv. Nanicão) fruits. Own table based on da Mota et al. [59].

|  | Phosphorylase I | Phosphorylase II |
|---|---|---|
| KCl concentration for extraction | 0.125–0.2 M | 0.25–0.35 M |
| Subunit size [kDa] | 90.6 | 112 |
| Native molecular weight [kDa] | 155 | 290 |
| Affinity | High affinity towards branched glucans | Low affinity towards branched glucans; the highest affinity and specificity for maltopentaose and maltohexaose |
| Form | Dimer | Dimer |
| Isoelectric point [pH] | 5.0 | N/E |
| Other characteristics | Similar to the cytosolic phosphorylases from other plant tissues | N/E |
|  | N/E | Has an ability to attack starch granules when associated with a-amylases |

Carbohydrates present in banana BPs can be fermented to obtain lactic acid [60]. Lactic acid is mainly produced by lactic-acid-bacteria-type species *Lactobacillus delbrueckii*. *Lactobacillus delbrueckii* are important because of their significant resistance to food processing conditions, like a high concentration of sodium chloride (up to 6%) and storage at low temperatures (4 °C) for 28 days, and notable tolerance for acid and bile salt conditions [61]. *Lactobacillus delbrueckii* produce very high amounts of lactic acid when enhanced with bioaugmentation and the enzymatic hydrolysis of substrates for microbiological mediums—0.65 g/$g_{total\ sugars}$ and 0.72 g/$g_{total\ sugars}$, respectively. However, simple pH standardization to 6.2 alone yields a significant proportion of lactic acid—0.57 g/$g_{total\ sugars}$. *Lactobacillus delbrueckii* can efficiently convert a variety of raw materials, including agro-industrial wastes like banana BPs, into lactic acid. This not only helps in reducing production costs but also aids in waste management. The utilization of household bio-waste and other low-cost substrates for lactic acid production has been demonstrated to be effective and sustainable [62]. Agricultural wastes, including banana residues, contain a variety of nutrients that are beneficial for microbial growth. This can lead to higher yields and productivity of lactic acid during fermentation processes. The presence of both hexose and pentose sugars in these residues can be effectively utilized by genetically modified microorganisms, enhancing the overall efficiency of lactic acid production [63,64]. The production of lactic acid from banana peel, including the exact steps and waste products of this process, is shown in Figures 9 and 10. To provide a clear view and better understanding, the processes have been presented as graphs. Banana peels were used as a source for reducing sugar production, yielding about 18 g per 1 L of enzyme cocktail. Next, the reducing sugars were used as a carbon source for *Lactobacillus delbrueckii* lactic acid bacteria, providing 28 g of lactic acid per 1 L of inoculum mixture. The process was described as very effective, suggesting that banana BPs are a valuable ingredient for lactic acid production [65].

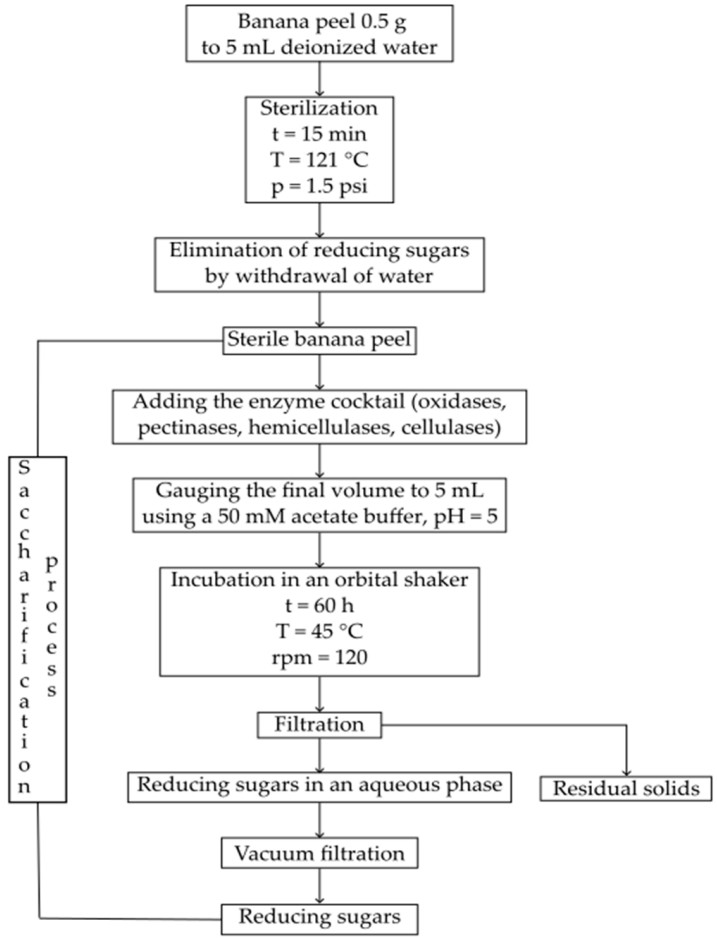

**Figure 9.** Extraction of reducing sugars from banana peel as an essential step for the lactic acid production process. Own figure based on Martínez-Trujillo et al. [65].

The pseudostem is also abundant in polyphenols and flavonoids including ferulic acid—a valuable antioxidant [66]—gentisic acid, (+)-catechin, protocatechuic acid, caffeic acid, and cinnamic acid [67]. The mentioned compounds exhibit antimicrobial, antioxidative, neuroprotective, chemopreventive, anticancer, and antiproliferative properties [4]. Present glucooligosaccharides and D-allulose boost the growth of probiotic bacteria. Additionally, D-alluloze has very few calories [68]. Ethanol extracts from the pseudostem exhibit antihyperglycemic properties in rats [69].

Cavendish banana leaves contain a membrane-bound enzyme of 9-LOX, which can be used to obtain oolong tea, melon, and fruity resembling flavors if pickled or treated with soybean oil or linoleic acid [70].

The highest concentration of phenolics is in the raw peel; thus, the peels should be processed fresh [71]. The peel is also a notable source of pectin [72], saponins, sterols, triterpenes (β-sitosterol, stigmasterol, campesterol, cycloeucalenol, cycloartenol, 24-methylene cycloartanol) [73], aromas like isoamyl acetate which is mostly used in banana flavoring [74], flour [75], and banana oil (amyl acetate) [8]. Peel extracts can be used as a reducing and stabilizing agent in the green acquisition of silver [76] and gold nanoparticles [77]. Recovering such valuable compounds may benefit developing countries with an additional source of income, where poverty is an issue.

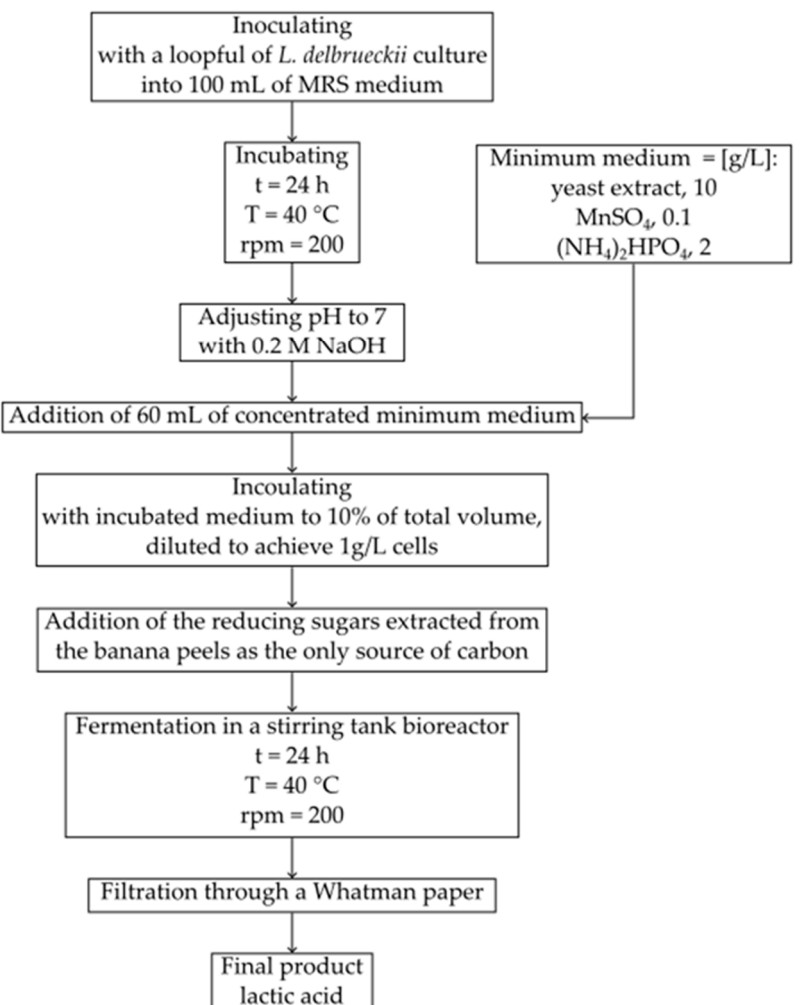

**Figure 10.** Lactic acid production process from reducing sugars extracted from banana peels. Own figure based on Martínez-Trujillo et al. [65].

*2.4. Snacks*

Banana plant parts may be incorporated into dessert products and sweets to increase their sensory and functional properties. The inflorescence may be mixed with gram, wheat, potato, corn flour, and cocoa powder to obtain various snacks like laddu, plain cakes, nut chocolate, nuggets, biscuits, or dark chocolate [78]. Banana flower water extracts can be used to brew a soft drink rich in phenolic compounds, which was proven beneficial for breastfeeding, postpartum, and post-cesarean women [79].

A notable amount of banana fruit is lost during the manufacturing and post-harvesting processes. To prevent this, unripe bananas can be processed into much more stable flour. Green banana flour contains enough antioxidants and essential minerals to fortify deficient food products [80,81]. This is used to enrich cassava crackers and fish crackers, which are considered poor nutrition-wise [82].

Banana peels may be incorporated into many snacks like gluten-free cakes, waffle cones, gluten-free rissoles, biscuits, cereal bars, and jelly. Doing so increases the fiber, polyphenol, and antioxidant content without a notable impact on the texture, flavor, aroma, and overall acceptability [83].

The male buds, pseudostem, and unripe fruit can be used in the production of chips, figs, instant drinks, flour, jam, candy, dehydrated slices, and preserves [8].

## 2.5. Fuel

Banana plant pseudostem and leaves can be used to produce fuel briquettes. Such biomass fuel is cheap to produce and provides about 17.10 MJ/kg of energy regarding leaves and 13.70 MJ/kg for the pseudostem. This solution seems advantageous for India, Indonesia, Thailand, Pakistan, Ghana, and South Africa, where fossil fuels are not easily available for the poorest people or are expensive [84]. The downside of such biomass pellet production is the emission of greenhouse gasses, e.g., methane; however, methane with other produced gasses (CO, $H_2$) can be used in further power generation [6].

The pseudostem may be used to produce hydrogen and bioethanol due to a significant carbohydrate content [85].

Considering the high amount of cellulose, banana peels make for a cheap bio-substrate for fuels, organic acids, enzymes, sugars, ethanol, xylanases, laccase, and xylitol and can be used as a base for edible mushroom cultivation [9]. Moreover, carbonized peel may be utilized as hierarchically porous activated carbon scaffolds for energy storage applications, either as high-performance activated banana peel/$NiCo_2O_4$ electrode materials for supercapacitor applications or as an activated banana peel/ Ni/graphene composite for Li/S batteries [86].

The combination of many BPs, mainly the peel, pseudostem, and leaves, shows promising results in manufacturing briquettes for heating and energy generation. Such fuel provides about 12–22 MJ/kg of energy, meaning impressive combustibility. In this case, banana peels act not only as a combustible polysaccharide source but also as a binder for the whole briquette, making it easy to mold [86].

Another way of providing energy is the use of microbial fuel cells (MFCs). An MFC is a bio-electrical device capable of transforming chemical energy provided by organic matter into electrical energy using catalytic reactions performed by microbes. An MFC may be utilized as an alternative to fossil fuels. According to Singh et al. [87], banana (*Musa acuminata*) peel slurry may be utilized as a substrate for MFCs, generating a maximum voltage of 0.20 V and a current of 0.112 mA regarding potassium dichromate catholyte and 0.42 V and 0.55 mA, respectively, regarding potassium ferricynide catholyte. Utilizing banana peels in MFCs helps to reduce waste and provides an eco-friendly and cost-effective method for energy production. MFCs powered by banana peel waste are an emerging technology for bio-electricity production [88].

Using banana BPs as fuel helps to reduce the need to burn wood and therefore reduces deforestation, which benefits natural ecosystems [4].

## 2.6. Traditional Medicine and Disease Prevention

Peels can be used as medicine to treat dysentery [89], burns, anemia, diarrhea, inflammation, diabetes, cough, snakebite, and excess menstruation [9]. The consumption of banana peel helps with healing stomach ulcers, thickens the mucous membrane layer in the stomach [90], helps in overcoming depression due to the presence of tryptophan, mitigates anemia because of its high iron content, and stabilizes blood pressure due to a significant potassium content and a low sodium chloride content [89]. Moreover, banana peels are abundant in fructooligosaccharides, which stimulate beneficial bacterial growth in human intestines [91]. Unripe banana peel can be eaten to treat diarrhea [89] and vomiting [92]. Such properties of banana peel seem useful in, e.g., Tanzania, where some medicine is counterfeited and access to drugs is limited, especially in remote areas [93], or in China, Brazil, Venezuela, Iran, Iraq, and Jordan, where medicine shortages have been reported [94]. Moreover, consuming banana peel is safe regarding anti-nutritive compounds including oxalate, phytate, and hydrogen cyanide, whose level is below the permissible limit of 0.5–3.5 mg/g [9].

Banana peels contain dietary fiber and phenolic compounds, which exhibit antioxidant, antimicrobial, and antibiotic properties [9]. Phenolics are beneficial for human health, preventing cardiovascular disease, cancer, diabetes, and obesity [95,96], and are used to inhibit lipid oxidation and fungal and bacterial growth in food [97]. Banana peels

contain 5–47 mg gallic acid equivalent/g dry matter (mg GAE/g DM) phenolics [98,99], which is 1.5–3 times more than in the flesh [100]. Another phenolic compound found in banana peels—caffeic acid [101]—at a concentration of 0.2 mg/mL is capable of completely inhibiting the growth of *Aspergillus flavus* and *Aspergillus parasiticus* fungi, which produce aflatoxin [102]. There is a high rate of aflatoxin food contamination in developing countries, especially in sub-Saharan Africa. Aflatoxin poisoning is dangerous not only to locals but also to people who benefit from the imported food [103]; therefore, extracting the caffeic acid seems reasonable there.

### 2.7. Bio-Sorbents

Banana peels have been proven to work well as a cheap (about 10 times cheaper than activated coal) sorbent for heavy metals (lead, chromium, cadmium, copper, zinc) [104–106], radioactive materials [107], fluoride, pesticides [108], nitrate [109], carbon, sodium, aluminum, calcium, magnesium, and potassium salts or cations [110].

Banana pith may be used to sorb lead, copper, chromium, and zinc ions from electroplating waste and synthetic solutions [111].

Banana plant stalk can be used to manufacture bio-sorbents like acid-activated banana stalk, base-activated banana stalk, and raw banana stalk, which are used to adsorb lead (III) from aqueous solutions [112].

Lignocellulosic biomass obtained from banana plants' bark, pseudostem, leaves, and peels can be used to sorb pollutants of gaseous form like $H_2S$, $N_2O$, $NO_2$, $NH_3$, $CO_2$, SO2, and VOCs (volatile organic compounds). Moreover, sorbents from various banana wastes are utilized in metal ion, pesticide, water-soluble radioactive nuclide, and inorganic anion removal [113].

This may be useful in Bangladesh, China, Mexico, Egypt, Cambodia, and India, where heavy metal water pollution makes drinking or cooking with the water unsafe [27].

### 2.8. Natural Preservatives and Antimicrobial Compounds

Ethanolic extracts of banana peels exhibit antimicrobial properties. Compounds extracted from peels from Nigeria suppressed *Bacillus* sp., *Escherichia coli*, *Pseudomonas* sp., *Klebsiella pneumoniae*, *Staphylococcus aureus*, and *Streptococcus* sp. [114]. Moreover, ethanolic extracts inhibit the growth of *Lactococcus garvieae* and other strains, which attack freshwater fish. This is potentially useful in prawn disease prevention [115] in developing countries like India, where prawn farmers struggle with prawn bacterial diseases [116]. Ethyl acetate and ethanol extracts obtained from peels from Bangladesh suppressed Gram-positive bacteria—*Bacillus subtilis*, *Bacillus megaterium*, *Staphylococcus aureus*, and *Sarcina lutea*—and Gram-negative bacteria—*Salmonella paratyphi*, *Pseudomonas aeruginosa*, *Shigella boydi*, and *Vibrio mimicus* [117]. The preservative potential of the mentioned extracts has been proven to be similar to synthetic preservatives like butylated hydroxytoluene (BHT) and butylated hydroxyanisole (BHA) [118,119]. Despite food spoilage not being a significant issue in the rural areas of developing countries, people in big cities in, e.g., India, Pakistan, and Bangladesh struggle to keep food fresh because of limited access to power sources like fossil fuels [120]. Moreover, producing natural preservatives may be a novel source of income for the mentioned countries.

## 3. Banana By-Product Utilization Disadvantages

There are some disadvantages that limit the utilization of banana BPs. Fresh BPs have a significant moisture content often exceeding 80%, which makes them prone to spoiling, and thus require moisture elimination procedures and the use of preservatives, causing transportation, storage, and manufacturing costs to increase [49]. To partially overcome this disadvantage, it is suggested to use the peel of the ripest bananas, as water migrates from the peel to the fruit as ripening occurs [8].

Regarding animal feed, the processing costs of the BPs may even exceed the value of the final product; thus, BPs can be incorporated into other dry fodder ingredients to

reduce the production costs [121,122]. Another method of reducing the moisture content is microbial processing. Vacuum-drying is used to prevent nutrient loss while reducing moisture. A combination of both methods has also been suggested [123].

Given the abundance of the banana species, it is easy to assign some unique properties to all bananas, which is incorrect. Many works have shown different results regarding the chemical compounds in bananas [114]; thus, local research should be carried out to identify unique properties. This, however, may be costly.

Each time a washing process occurs in a technology utilizing banana and plantain BPs, a large amount of wastewater is produced—e.g., about 100 L per kg of plantain peel starch. This, however, can be managed by physicochemical treatment with aluminum salts—$Al_2O_3$ and $Al_2(SO_4)_3$—combined with coagulation, flocculation, and decanting. This procedure not only provides another useful BP—cellulose—but also reduces the water consumption (treated water recycling) and reduces wastewater effluent volume [20].

## 4. Banana By-Products as Fibers

One of the most useful banana BPs is banana fiber. The composition of banana fiber is presented in Table 4. The most important characteristics of banana fiber include the following: high durability (average strength of 3.93 cN/dtex), 3% elongation, light weight, medial length of 50~60 mm, medial fineness of 2386 Nm, ability to absorb and release water freely, ability to be spun in various ways like ring spinning, open-end spinning, bast fiber spinning, and semi-worsted spinning, and biodegradability. Banana fiber mechanical properties are presented in Table 5. The banana fiber can be produced manually, giving a rather low yield of 500–600 g per 8 h per worker. To increase the yield, machine technology can be introduced. The idea of parallel banana farming and BP management was introduced for Bangladesh by Mohiuddin et al. [124]. This may be especially important in developing countries which plan to start banana plantations and will inevitably face BP management [124].

**Table 4.** Composition of banana fiber. Own table based on [124].

| Compound | Content, % |
|---|---|
| Cellulose | 50–60 |
| Hemicelluloses | 25–30 |
| Lignin | 12–18 |
| Pectin | 3–5 |
| Water-soluble materials | 2–3 |
| Fat and wax | 3–5 |
| Ash | 1–1.5 |

**Table 5.** Mechanical properties of banana fiber. Own table based on [125,126].

| Mechanical Property | Value |
|---|---|
| Density | 1–1.5 g/m$^3$ |
| Elongation at break | 4.5–6.5% |
| Young's modulus | 20 GPa |
| Microfibrillar angle | 11 deg |
| Lumen size | 5 mm |
| Average strength | 3.93 cN/dtex |
| Medial fineness | 2386 Nm |

Banana pseudostem is a source of natural fiber consisting mainly of cellulose but also of hemicellulose, lignin, and pectin. The fiber is located in the sheaths of the pseudostem. Out of the 14–18 sheaths present in the pseudostem, the inner 4–6 sheaths make the softest fibers, the middle 6–8 sheaths constitute soft, lustrous fibers, and the outer 4–6 sheaths make coarse fibers. Banana fiber exhibits satisfactory mechanical properties, similar to, e.g., glass fiber. It is the strongest natural fiber, light in weight, resistant to combustion,

and biodegradable. Despite its assets, raw banana fiber is of little use. Most often it is woven into more complex structures like yarn and then ropes, fabric, and finally clothes and apparel, or it is a part of composites. It is used as a compound for making lamp stands, car underfloor protection panels, or banknotes [127].

Banana fibers are used to produce woven composites. One of these is a composite of natural banana fiber and epoxy. The banana epoxy composite was capable of withstanding stress in the $x$-direction up to 14.14 MN/m$^2$ and in the $y$-direction up to 3.398 MN/m$^2$. Considering the Young's modulus, the obtained values were 0.976 GN/m$^2$ and 0.863 GN/m$^2$ for the $x$ and $y$-directions, respectively. To bend the composite beam by 0.5 mm, the maximum required load was 36.25 N considering the three-point bend test. The banana epoxy composite shows high stability during various tests. This is expected to be useful in Malaysia, since a lot of village residents use woven banana fibers as a resource in producing household utilities [12].

Banana fiber has many uses, especially in countries where bananas are grown. The majority of banana fiber is utilized to craft ropes and cordage due to the fiber's high resistance to sea water and its innate buoyancy. Other uses include wall drilling cables, fishing nets, lines, mats, composite materials, packaging, eco-friendly fabrics, sheets, fine cushion covers, neckties, bags, table cloths, curtains, rugs, sanitary napkins, paper pulp, sacks, handbags, textiles, baskets, wall hangings, floor mats, decorative papers, and socks [8,128–130]. Banana fiber is particularly useful in textile production. It is resistant to alkali, chloroform, acetone, formic acid, phenol, and petroleum ether. In comparison with other fabrics like cotton or chemical fibers, banana fiber is lustrous and exhibits significant water absorption. Moreover, it rarely causes any irritation to skin and is environmentally friendly. In Japan, it is used in traditional outfits like the kimono and kamishimo [131].

Bio-fibers like banana fiber have some limitations. Due to their high moisture absorption ability, it is often difficult to combine them with hydrophilic polymers [125]. This however can be overcome by pH and temperature treatment and pulping. Banana fiber paper made this way was reported to exhibit lower water absorption than wood pulp paper [132]. Another way to produce paper with banana fiber is to soak the fiber in water, making pulp. Then, the pulp is bleached using *Trichoderma* and *Pythium* fungi treatment for 3–5 days. These fungi break bonds between lingo-cellulosic complexes, causing lignin and hemicelluloses to be removed from the matrix. This whole process contributes to the increased brightness and softness of the final product and makes pulping easier. Next, the pulp is washed to remove fungi, making it ready for the main step—beating. The last step is to add any necessary additives, which enhance the strength and texture and make the paper less prone to wetting and puncturing. The additives include starches, polysaccharide resins, and natural gums. This paper is mainly used for manufacturing shopping bags, file sheets, visiting cards, greeting cards, invitation covers, scribing pads, envelopes, art paper, or printing paper [112]. This is especially useful in countries where wood paper is limited and banana BP pollution is notable, like Pakistan. Pakistan faces a problem with wood shortages due to very little forest areas and progressing deforestation. Since Pakistan produces a notable number of bananas, it seems reasonable to replace wood paper with banana BP paper [133]. Photographs of selected products made of banana fiber are presented in Figures 11–13.

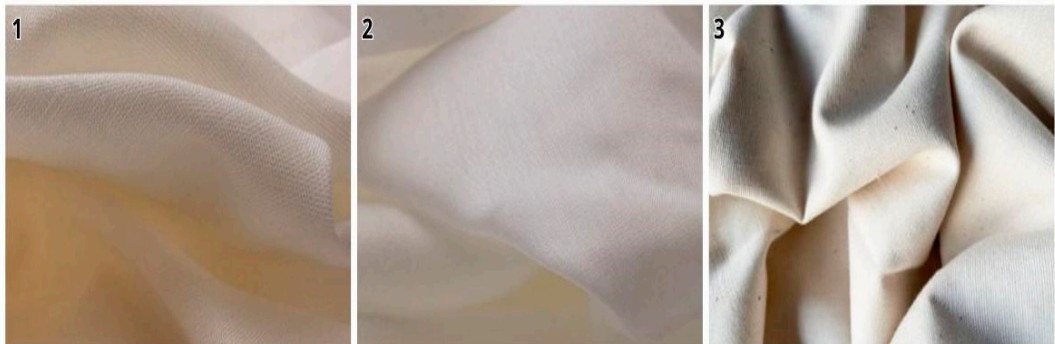

**Figure 11.** Fabric made of banana palm fiber. (**1**) Banana fabric made using a handloom and (**2**) a powerloom. (**3**) Banana fabric mixed with organic cotton. Reproduced from [134] and modified with kind permission from (Agriculture Information).

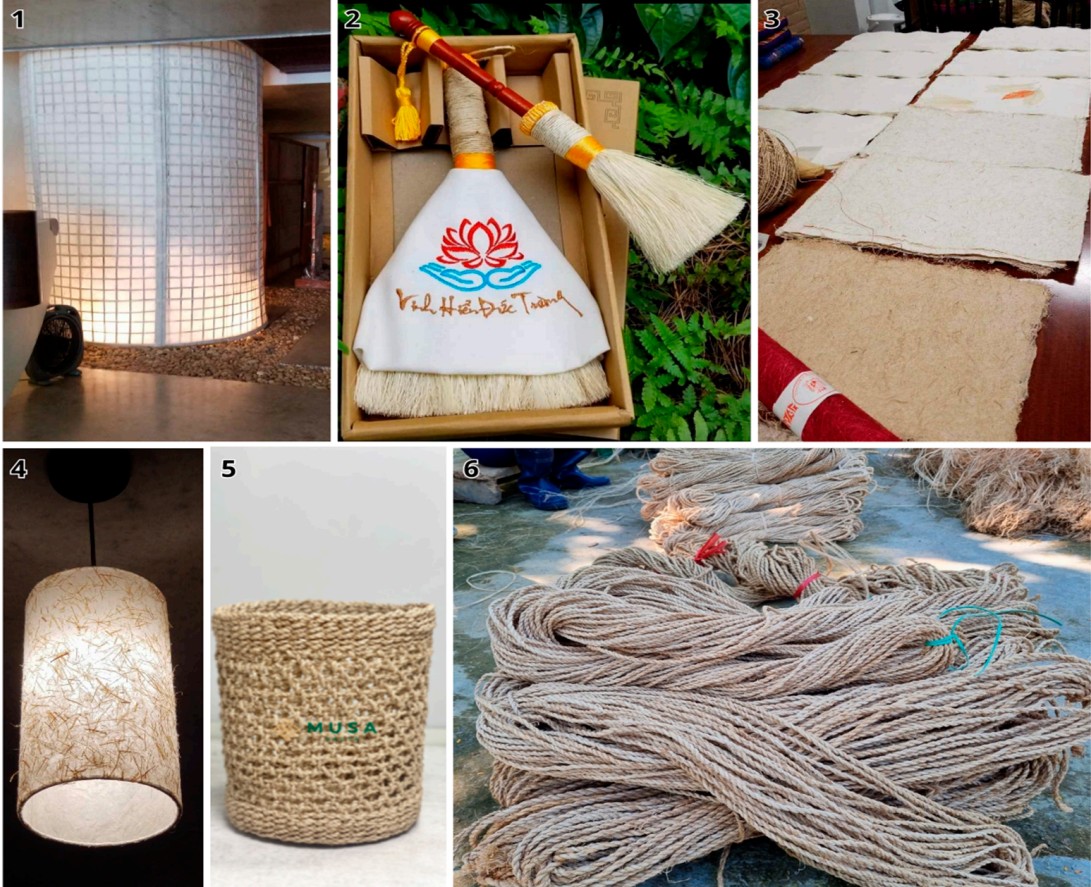

**Figure 12.** Various products made of banana fiber—(**1**) indoor decoration, (**2**) a cleaning brush, (**3**) paper, (**4**) a lamp, (**5**) a basket, and (**6**) ropes. Reproduced from [135] and modified with kind permission from Ecosilky.

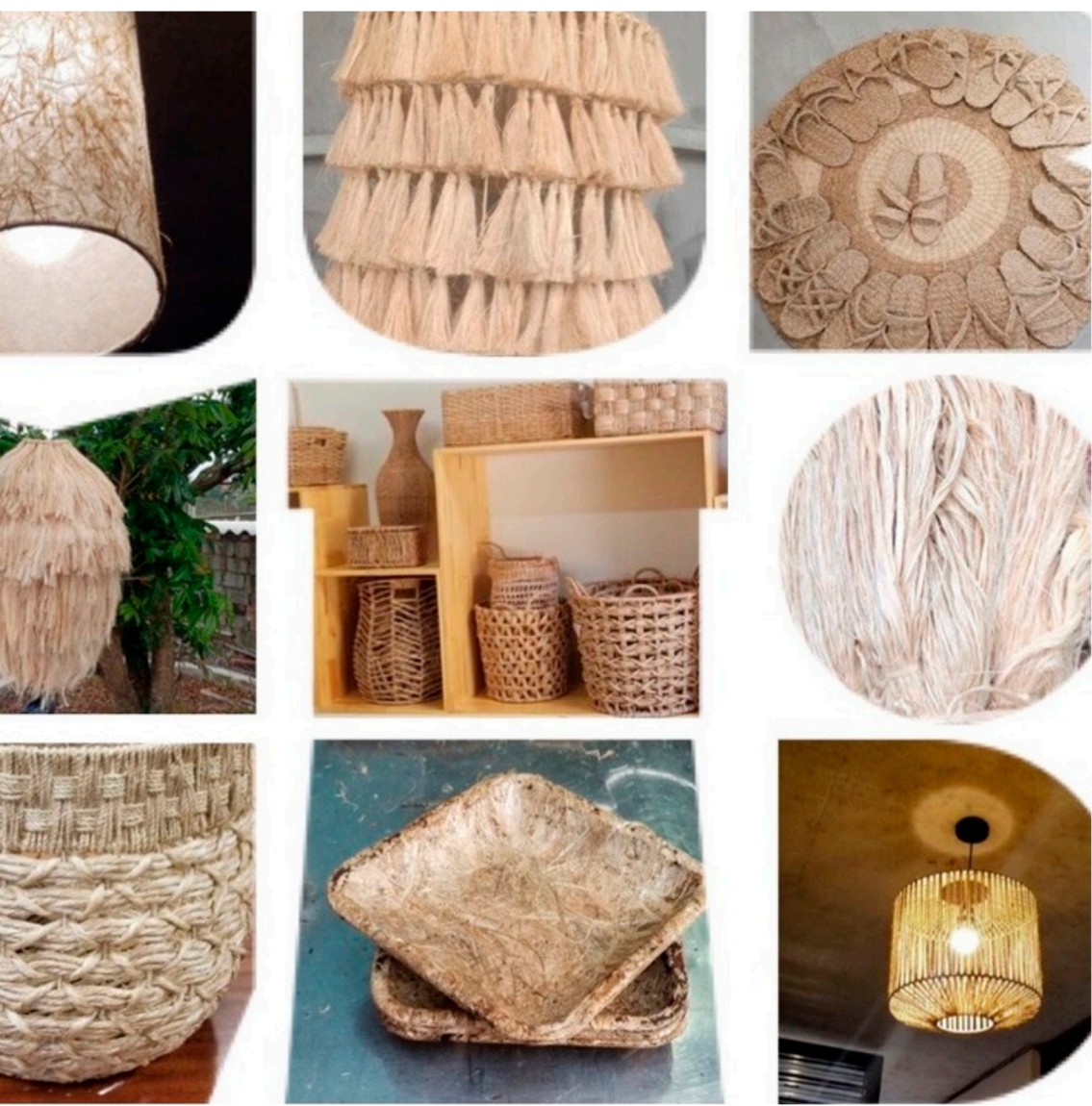

**Figure 13.** Various banana fiber products. Reproduced from [136] and modified with kind permission from Musa Pacta.

Unripe plantain peels may be used for nanofiber production. Tibolla et al. [137] introduced a composite made with cellulose nanofibers from discarded banana peels. The diameter and length were 10.9 nm and 454.9 nm for the chemical process and 7.6 nm and 2889.7 nm for the enzymatic process, respectively. This is a way to utilize unripe fruit in Brazil, which are discarded as waste [137].

### 5. Conclusions

Bananas and plantains are among the most important fruits worldwide. Enormous production comes with a great deal of BPs often discarded as waste. Banana BPs are mostly a source of natural polysaccharide polymers such as gums, lignin, cellulose, hemicellulose, lignocellulose, or fiber. These polymers are used to manufacture bio-sorbents, bio-fuel, briquettes, or biogas, in biodegradable fabrics and everyday items, and to enrich deficient food in dietary fiber. However, other BPs can be used to overcome many social, health, ecological, and financial issues present in developing countries. The significant obstacles to complete BP utilization are the shortage of money, unqualified staff, and the lack of technical background. This leads to a situation where the insufficiency of means causes the

piling up unutilized BPs, while unutilized BPs may be a solution to overcome the lack of means. Given this, fair external investment might be a solution to the presented problem, helping developing countries to develop.

The future of banana BPs shows promise. There are many opportunities to consider in the upcoming years. Due to global warming, the climate of some areas will change. These changes can be predicted, and, in some cases, new plant species may be required to fight deforestation and desertification. One such plant may be the banana palm due to its high-temperature requirements. Moreover, banana palms provide shade and shelter for other plants and small animals.

Another novel application for banana BPs may be incorporating banana fiber into modern clothing. Since people worldwide are more and more concerned with renewable and eco-friendly solutions including apparel, the utilization of biodegradable banana fiber may interest clients and manufacturers. This may also increase the demand for banana BPs, giving jobs to people in developing countries and reducing banana BP pollution. Moreover, banana fiber is a great component in innovative composites. It may be possible to develop fabrics with new properties, exceptional strength, or resistance to crumpling.

Since banana leaves and inflorescences contain aroma compounds, it may be worth considering bio-elicitation. Using bio-elicitors may increase the amount of aroma and flavor compounds extracted from the leaves and inflorescences, making the process more efficient and profitable.

Banana peel is a valuable source of polysaccharides, which may be used for the production of reducing sugars. Reducing sugars can then be used for the production of other compounds like lactic acid. Given this, it seems reasonable to use other banana palm BPs like the pseudostem or unripe fruit as a source of reducing sugars, since these BPs contain high amounts of polysaccharides.

**Author Contributions:** Conceptualization, R.W.; investigation, R.W.; writing—original draft preparation, R.W.; writing—review and editing, B.G.S.; visualization, R.W.; supervision, B.G.S.; funding acquisition, B.G.S. All authors have read and agreed to the published version of the manuscript.

**Funding:** This research received no external funding.

**Institutional Review Board Statement:** Not applicable.

**Data Availability Statement:** No new data were created.

**Conflicts of Interest:** The authors declare no conflicts of interest.

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
