# Peer review of "Potential Valorization of Banana Production Waste in Developing Countries: Bio-Engineering Aspects"

_fibers, doi:10.3390/fib12090072_

Round 1
Reviewer 1 Report
Comments and Suggestions for Authors
The submitted manuscript is a very interesting review about banana byproducts utilization. Provides very interesting information on the examined issue. However, I have some suggestions for improvement and overall clarity.
L33-35: in which country or region? Probably you mean worlwide.
L36-36: why this abnormal situation is happening? Is there any explanation? If yes, please mention it, it could be interesting for the readers.
Table 2: refer what %, you mean. Probably % w/w, also add some standard deviations of your values. If these values are mean values based on data previously presented in the literature, I suggest to present them in the form of intervals (ranging from the lowest to the highest value).
Moreover, a standard deviation with a 0.00 value is not possible. Also Iron (Fe) is missing standard deviation value.
Some potential suggestions:
Usage of banana flower in the agriculture and food industry.
I suggest a minor revision before acceptance.
Author Response
Response to Reviewer 1 Comments (in yellow)
|
||
1. Summary |
|
|
Thank you very much for taking the time to review this manuscript. Please find the detailed responses below and the corresponding revisions/corrections highlighted/in track changes in the re-submitted files. |
||
2. Point-by-point response to Comments and Suggestions for Authors |
|
|
Comment 1: L33-35: in which country or region? Probably you mean worlwide. |
||
Response 1: Thank you for pointing this out. We agree with this comment. Therefore, we have provided with missing word. L33. |
||
Comment 2: L36-36: why this abnormal situation is happening? Is there any explanation? If yes, please mention it, it could be interesting for the readers. |
||
Response 2: Thank you for pointing this out. We agree with this comment. Therefore, we have provided with the explanation. L36-L40.
Comment 3: Table 2: refer what %, you mean. Probably % w/w, also add some standard deviations of your values. If these values are mean values based on data previously presented in the literature, I suggest to present them in the form of intervals (ranging from the lowest to the highest value). |
Response 3: Thank you for pointing this out. We agree with this comment. Therefore, we changed the information in the table accordingly. Table 2.
Comment 4: Some potential suggestions: Usage of banana flower in the agriculture and food industry.
Response 4: Thank you for pointing this out. We agree with this comment. Therefore, we extended the content about banana flower utilization. L175-180

Reviewer 2 Report
Comments and Suggestions for Authors
Brief summary:
The authors present the review paper "Potential Valorization of Banana Production Waste in Developing Countries: Bio-Engineering Aspects". The article highlights the advantages of using banana by-products (BPs) to address problems in developing countries. The authors begin by explaining the global importance of banana harvests and why banana waste is a concern. They discuss diverse banana by-products such as peels, leaves, pseudostems, etc., and their industrial applications as fuels, medicine, biosorbents, and other uses. The review also explores the use of banana by-products as fibers, for example, in fabrics, paper, and other applications. The authors goal is to generate ideas that motivate the research of new knowledge that will inspire the innovation of new technologies.
In addition, the use of banana fibers for agro-industrial applications is very important, the paper could add more information about this. Overall, this paper presents relevant information that could inspire new technologies. In developing countries, this could be a source of new jobs, addressing economic issues in these nations.
Specific comments:
Here are some observations that need attention:
Line 42. The acronym BPs (by-products) is defined in the Abstract, but mentioning it again here could help readers identify it more easily.
52-54. The article referenced in this phrase does not discuss carbohydrates from bananas. The phrase was extracted from the article, but it cites other articles dating back to 2006 (see https://www.sciencedirect.com/science/article/pii/S0308814617308191?via%3Dihub). I suggest updating this reference with more recent sources. If discussing carbohydrates from bananas specifically, include relevant references; otherwise, clarify that these are properties of carbohydrates in general.
57. I strongly recommend improving the image resolution and changing the symbology used, as it's difficult to identify the areas. Using colors could facilitate reading.
67. What does the 'm' at the end of the data refer to?
110-129. More references about cosmetic use are required. Please consult additional sources for these claims.
150. Using schemes to explain the method is a good tool for reader orientation. However, these schemes require more explanation or better justification for their use.
232. Table 3 shows good data about phosphorylase from unripe bananas, but more explanation about its implications and importance is necessary.
238. The authors state that reducing sugars can be a carbon source for Lactobacillus delbrueckii, but more text about the importance of this is necessary. For example, discuss the industrial impact of this microbial species and why using banana by-products as a carbon source is important.
309. Fuel application is interesting data. I recommend adding more waste-valorization technologies, for example, microbial fuel cells.
311. I suggest placing the reference number at the end of the sentence.
320. Replace "Brazin" with "Brazil".
323. References are necessary for this claim. Please add the relevant references.
328. Please replace "garlic" with "gallic".
410-412. Data and references for this claim are required. Please add data and references at the end of the sentences.
Regarding the use of banana by-products as fibers, I suggest incorporating a table that presents their mechanical properties.
Author Response
Response to Reviewer 2 Comments (in green) |
||
1. Summary |
|
|
Thank you very much for taking the time to review this manuscript. Please find the detailed responses below and the corresponding revisions/corrections highlighted/in track changes in the re-submitted files.
|
||
2. Point-by-point response to Comments and Suggestions for Authors |
|
|
Comment 1: Line 42. The acronym BPs (by-products) is defined in the Abstract, but mentioning it again here could help readers identify it more easily
|
||
Response 1: Thank you for pointing this out. We agree with this comment. Therefore, we have added the explanation. L46
|
||
Comment 2: 52-54. The article referenced in this phrase does not discuss carbohydrates from bananas. The phrase was extracted from the article, but it cites other articles dating back to 2006 (see https://www.sciencedirect.com/science/article/pii/S0308814617308191?via%3Dihub). I suggest updating this reference with more recent sources. If discussing carbohydrates from bananas specifically, include relevant references; otherwise, clarify that these are properties of carbohydrates in general. |
||
Response 2: Thank you for pointing this out. We agree with this comment. Therefore, we have added recent source and diversified general and more specific information. L56-61
Comment 3: 57. I strongly recommend improving the image resolution and changing the symbology used, as it's difficult to identify the areas. Using colors could facilitate reading. |
Response 3: Thank you for pointing this out. We agree with this comment. Therefore, we improved the image’s quality. L64
Comment 4: 67. What does the 'm' at the end of the data refer to?
Response 4: Thank you for pointing this out. The “m” is explained in the table caption as “m expressed as % molar proportion”. L73
Comments 5: 110-129. More references about cosmetic use are required. Please consult additional sources for these claims.
Response 5: Thank you for pointing this out. We agree with this comment. Therefore, we added additional information. L147-148
Comment 6: 150. Using schemes to explain the method is a good tool for reader orientation. However, these schemes require more explanation or better justification for their use.
Response 6: Thank you for pointing this out. We agree with this comment. Therefore, we justified usage of the schemes. L142-147; 243-252; 266-285
Comment 7: 232. Table 3 shows good data about phosphorylase from unripe bananas, but more explanation about its implications and importance is necessary.
Response 7: Thank you for pointing this out. We agree with this comment. Therefore, we provided with additional information. L243-250
Comment 8: 238. The authors state that reducing sugars can be a carbon source for Lactobacillus delbrueckii, but more text about the importance of this is necessary. For example, discuss the industrial impact of this microbial species and why using banana by-products as a carbon source is important.
Response 8: Thank you for pointing this out. We agree with this comment. Therefore, we added more information. L266-282
Comment 9: 309. Fuel application is interesting data. I recommend adding more waste-valorization technologies, for example, microbial fuel cells.
Response 9: Thank you for pointing this out. We agree with this comment. Therefore, we added information about microbial fuel cells. L356-365
Comment 10: 311. I suggest placing the reference number at the end of the sentence.
Response 10: Thank you for pointing this out. We agree with this comment. Therefore, we placed the reference number accordingly. L367
Comment 11: 320. Replace "Brazin" with "Brazil".
Response 11: Thank you for pointing this out. We agree with this comment. Therefore, we replaced the wrong word. L378
Comment 12: 323. References are necessary for this claim. Please add the relevant references.
Response 12: Thank you for pointing this out. We agree with this comment. Therefore, we added the reference. L382
Comment 13: 328. Please replace "garlic" with "gallic".
Response 13: Thank you for pointing this out. We agree with this comment. Therefore, we replaced the wrong word. L387
Comment 14: 410-412. Data and references for this claim are required. Please add data and references at the end of the sentences.
Response 14: Thank you for pointing this out. We agree with this comment. Therefore, we added the references. L451
Comment 15: Regarding the use of banana by-products as fibers, I suggest incorporating a table that presents their mechanical properties.
Response 15: Thank you for pointing this out. We agree with this comment. Therefore, we provided the table. L458-459; 467-468.
